# Solving Monocular Visual Odometry Scale Factor with Adaptive Step Length Estimates for Pedestrians Using Handheld Devices

**DOI:** 10.3390/s19040953

**Published:** 2019-02-23

**Authors:** Nicolas Antigny, Hideaki Uchiyama, Myriam Servières, Valérie Renaudin, Diego Thomas, Rin-ichiro Taniguchi

**Affiliations:** 1Institut Français des Sciences et Technologies des Transports, de l’Aménagement et des Réseaux (IFSTTAR) AME GEOLOC, 44340 Bouguenais, France; valerie.renaudin@ifsttar.fr; 2Centrale Nantes, 44300 Nantes, France; myriam.servieres@ec-nantes.fr; 3Institut de Recherche en Sciences et Techniques de la Ville (IRSTV), 44300 Nantes, France; 4Laboratory for Image and Media Understanding (LIMU), Kyushu University, Fukuoka 819-0395, Japan; uchiyama@limu.ait.kyushu-u.ac.jp (H.U.); thomas@ait.kyushu-u.ac.jp (D.T.); rin@kyudai.jp (R.-i.T.); 5Centre de Recherche Nantais Architectures Urbanités (CRENAU) AAU, 44262 Nantes, France

**Keywords:** pose estimation, localization, handheld device, pedestrian navigation, urban mobility, augmented reality

## Abstract

The urban environments represent challenging areas for handheld device pose estimation (i.e., 3D position and 3D orientation) in large displacements. It is even more challenging with low-cost sensors and computational resources that are available in pedestrian mobile devices (i.e., monocular camera and Inertial Measurement Unit). To address these challenges, we propose a continuous pose estimation based on monocular Visual Odometry. To solve the scale ambiguity and suppress the scale drift, an adaptive pedestrian step lengths estimation is used for the displacements on the horizontal plane. To complete the estimation, a handheld equipment height model, with respect to the Digital Terrain Model contained in Geographical Information Systems, is used for the displacement on the vertical axis. In addition, an accurate pose estimation based on the recognition of known objects is punctually used to correct the pose estimate and reset the monocular Visual Odometry. To validate the benefit of our framework, experimental data have been collected on a 0.7 km pedestrian path in an urban environment for various people. Thus, the proposed solution allows to achieve a positioning error of 1.6–7.5% of the walked distance, and confirms the benefit of the use of an adaptive step length compared to the use of a fixed-step length.

## 1. Introduction

In the context of pedestrian navigation and handheld device pose estimation for urban mobility, localization and orientation would gain from an accurate global pose estimation using Global Navigation Satellite Systems (GNSS), Inertial Measurement Unit (IMU) and cameras [1,2,3]. Such systems generally operate well in outdoor open environments. However, urban environments comprising closely spaced buildings (i.e., urban canyon) still represent challenging areas for GNSS, which suffer from attenuation, reflection, and blockage effects [4,5,6]. It is even more challenging using low-cost Micro Electro Mechanical System (MEMS) sensors and low computational resources, which are typically embedded in pedestrian mobile devices [7]. Moreover, the hand can move freely even when the pedestrian’s centre of mass is static, making it difficult to estimate an accurate pose.

In order to assist the pedestrian navigation during large displacements in urban environments, e.g., through on-site augmented reality applications on handheld mobile device, our aim is to provide a continuous and accurate pose estimate. In such context, we propose to integrate human motion estimation, i.e., step length estimation, using a handheld IMU proposed in the field of Pedestrian Dead Reckoning (PDR), to scale the monocular Visual Odometry (VO). To complete the pose estimation, we propose to use an estimation of the holding in hand of mobile devices coupled with data content in Geographical Information System (GIS). Although embedded technologies in general public devices become more efficient, the choice was made to propose a solution that can be fully embedded with a minimal hardware setup and a low memory requirement, and without any connection with networks, any deployment of new infrastructures, to be totally autonomous.

Thus, our first contribution is to achieve a continuous pose estimation with a mobile device held in hand during large displacements in urban environments. Our second contribution is to present an adaptive method, based on the use of an IMU and a monocular camera, that takes advantage of human motion estimation and data content in Geographic Information Systems to dynamically solve the scale ambiguity and suppress the scale drift in the monocular Visual Odometry. Finally, our third contribution is to propose a solution that does not require pedestrians to make specific and unnatural movements or to revisit the same place, as is the case with Simultaneous Localization And Mapping (SLAM) techniques [8,9,10].

The Section 2 conducts a state of the art on pose estimation and Visual Odometry scale estimation. The Section 3 presents the coupling process between PDR and VO, i.e., the scale estimation based on step length estimation, and the position estimation on the vertical axis using a hand-held model of the equipment and GIS data. The Section 4 details the hardware setup used in our experiments, the acquisition scenario and the establishment of reference waypoints and tracks to evaluate our method. The assessment by comparison between a foot-mounted INS aided by GNSS phase measurements and the proposed method using a handheld device is presented in the Section 5. Finally, we conclude that human motion analysis can be informative as an additional clue and constraint to scale the monocular Visual Odometry and to improve the pose estimation with general public handheld devices.

## 2. Related Work

In this section, we briefly summarize the solutions for handheld device pose estimation in urban environments using a camera or/and an IMU. In the literature on urban localization, one approach is to use a camera and a database comprising images associated with their global poses, namely appearance-based localization [3,11,12]. The pose of the camera is determined by searching an image containing a part of the input image in the database. This is typically performed by using Content-Based Image Retrieval techniques. However, the accuracy of this approach strongly degrades under appearance changes. In addition, such a huge database is not always available, not easy to prepare, and also may not be applicable to mobile devices for pedestrians due to limited computational resources.

In the field of pedestrian navigation, Pedestrian Dead-Reckoning approaches have been investigated for the urban localization [13,14]. The pose is incrementally determined by computing the displacement between the poses after the initial global pose is acquired with other approaches such as appearance-based localization. Sensor fusion is an alternative approach and has also been investigated to compensate for the drawback of each sensor [15,16,17,18]. The fusion was initially introduced to compensate for weaknesses in appearance-based approaches due to failure under fast rotational motions (i.e., motion blur and large displacement in the images) [15,19,20]. A gyroscope in the IMU has an advantage that it can accurately track sensor angular rates at high frequency for short time intervals. An accelerometer in the IMU is useful during static phases to compensate for global orientation estimation by using the gravity measurement.

Monocular Visual Odometry is a frame-to-frame relative pose tracking approach using a camera for the incremental pose estimation [21,22,23]. In unknown environments, the camera pose is incrementally determined by, first, computing feature correspondences between two consecutive images, and then, computing the relative pose from the correspondences. However, one problem is to solve for scale ambiguity in the estimated poses [23]. The relative poses can fundamentally be estimated up-to-scale if known landmarks are used into the VO [8] or an initial translation of a camera is assumed to be known [24]. These approaches solve the initial scale estimation but still face the problem of scale drift that occurs in large displacements and degrades the localization accuracy [25]. As reported in [26], both direct and indirect VO approaches cannot avoid the scale drift when only a monocular camera is available, and the tracking errors are quickly accumulated over time. The drift can be punctually corrected by using appearance-based localization, using some known landmarks [3,27] or knowledge of existing 3D city models [28]. Ref. [29] presents a method for resolving the scale ambiguity and drift by using the slant distance obtained from a skyline matching between the camera and images synthesized using a 3D building model. The correction of the scale factor shows a 90% improvement of the positioning solution compared to a solution that does not correct the scale drift. These methods require to be located in an area where the 3D model is available and known with precision, whereas it is generally not. Loop closure, when revisiting the same place, is also used to correct the drift [25]. However, this approach is not applicable in the context of pedestrians navigating from point A to point B, without passing through the same place again.

Since the unit of the accelerometer in the IMU is metric, it is also useful to estimate the scale in the VO [30,31,32]. To estimate the scale, one approach is to compute the double integration of the acceleration after removing the gravity elements. However, the double integration is sensitive to noise because the error is quickly accumulated, even though the error is small during a short time interval. To further improve the accuracy and stability of metric scale estimation, additional methodologies need to be investigated. An alternative approach is to estimate the scale with an Extended Kalman Filter, that includes the scale factor in the state vector. As reported in [31], the method is also sensitive to a dynamic bias which is difficult to estimate.

Others scale estimation approaches exist but there is a hard constraint that the sensors need to be rigidly fixed to the pedestrian. For example, Ref. [33,34] have the sensors attached to a helmet. This constraint is not valid in the context of handheld AR.

As a recent approach dedicated to pedestrians, the pedestrian face is used as a known object to estimate the scale [32]. On a mobile device (e.g., a smartphone), two cameras are usually installed: a user-facing camera and a world-facing camera. In the context of handheld AR, the user-facing camera captures the user’s face, while performing the VO by using the world-facing camera. When a relative pose of the two cameras is calibrated, the metric scale for the VO can be computed from the face. This setup is reasonable for handheld devices, however, this approach requires that the pedestrian be static while the hand is moving. It should also be noted that VO using a stereo camera does not have these scale issues because the scale can be uniquely determined from the disparity between two cameras [26].

Other approaches, close to ours, use a pedometer and an average step length for scale estimation in monocular VO [35,36]. As the pedestrian has to avoid others pedestrians, bikers, cars, elements in the scene and to wait at pedestrian crossings, the step lengths are not constant during the walk and a average step length does not correspond to pedestrian movement in urban environments. This is why we propose to use a dynamic step length estimation to scale monocular VO.

## 3. Scaled Monocular Visual Odometry

In this paper, we propose a novel approach for pose estimation with sensors held in hand based on monocular Visual Odometry and Pedestrian Dead-Reckoning. A human motion analysis from inertial data, i.e., a step length estimation, is used to dynamically solve the scale ambiguity and suppress the scale drift. The overview of the proposed method is illustrated in Figure 1. The proposed method can be divided into three main processes: “PDR” part, “VO” part and “Coupling” part. Since the “PDR” part and “VO” part are independent, our approach can be referred as a loosely coupled approach. The variables needed to develop the proposed method are all detailed in their corresponding following parts.

The “PDR” part is dedicated to the processing of IMU measurements. The same origin is defined for the IMU sensors and the IMU measurements are given in the *Body* frame, labeled by *b*. We estimate pedestrian step lengths with sensors held in hand, while the pedestrian walks naturally [37,38]. The process outputs a step length $step_{k}$ in meter at every instant of step $t_{step_{k}}$. The details are given in Section 3.1.

The “VO” part is dedicated to monocular VO for relative pose estimation based tracking. The origin of the camera frame is the optical center of the camera and the visual measurements are given in the *Camera* frame, labeled by *c*. In this part, because the scale estimation is independent of the image processing, any VO or vSLAM method can be applied if it outputs a complete pose estimate [23,39]. The implemented Visual Odometry framework is detailed in Section 3.2.

The “Coupling” part is dedicated to the scale determination by fitting trajectories from the “PDR” part and the “VO” part. This part finally solves the scale ambiguity in monocular VO. The scaled pose in monocular VO is the final output of the proposed solution. The final pose estimate is given in the *Navigation* frame (i.e., the North-East-Down (NED) frame), labeled by *n*. In the proposed solution, we need to consider the frequency difference between sensors and the timing of the scale estimation. The details are given in Section 3.4.

### 3.1. Step Length Estimation Process

Classically, inertial signals are integrated according to a strap-down mechanization to compute the recursive positions of an IMU. This is possible only if the accumulated error caused by low-cost inertial sensors is frequently calibrated or reset, such as using Zero velocity UPdaTes (ZUPT) [14]. As a moment of zero velocity does not always occur in the context of hand-held device-based pedestrian navigation, ZUPT is not easily achievable. Also, the location of the pedestrian is normally represented as a pedestrian’s center of mass on a 2D map. This is obviously different from the location of the device held in hand, because the hand performs free motions even when the pedestrian’s center of mass is static. Therefore, instead of double-integrating the measured accelerations, a step length model is adopted to derive step lengths as pedestrian’s displacements for handheld devices. We summarize the procedure of pedestrian step length estimation proposed in [37,38]. It is claimed that this step length estimation is performed with a 2.5% up to 5% error on the walked distance. The different steps of the procedure are presented in Figure 2 and the notations are detailed hereafter.

To estimate step lengths, IMU signals acquired with sensors held in hand are analyzed as follows. A first motion classification is operated to determine the walking phases or the static phases of the pedestrian. A peak detection and a thresholding on the energy are applied on gyroscope signal $\omega_{gyro}$ and accelerometer signal $f_{acc}$ to determine the events $t_{step_{k}}$ when the pedestrian’s foot comes in contact with the ground at $k^{th}$ step. The step frequency is also computed as $f_{k}$. A second motion classification is operated by analyzing the variance of the IMU measurements to determine the device’s carrying mode (*Static* and *Walking* in *Texting* or *Swinging* mode) according to [38]. Then, a generic model is used to compute the step lengths $step_{k}$ according to [37]. It is based on the user’s height $h_{ped}$ and on a set of three parameters {a,b,c} trained on 12 subjects.(1)$$step_{k}=h_{ped}\times \left(a\times f_{k}+b\right)+c$$

### 3.2. Monocular Visual Odometry

Because step length estimation is independent of monocular Visual Odometry, any existing methods can be used. Here, we introduce our implementation based on the following standard procedure [22]. As a pre-processing, a standard camera calibration, using a checkerboard [40] and modeling the camera as a pinhole camera [41], is performed for a fixed resolution to express the camera’s coordinates in a normalized space [42]. This enables to correct image distortions and to determine the intrinsic parameters matrix K of the camera.

In the VO, to determine the unknown pose at time (t), the known poses at (t−1) and (t−2) are used. The different steps of the vision procedure are detailed in Figure 3 and the notations are detailed hereafter.

Knowing the intrinsic parameters matrix K of the camera and the correspondences between $x_{i}$ and $X_{n}$, expressed in homogeneous coordinates symbolized with a tilde (~), the pose of the camera’s optical center can be computed. It is expressed as a rotation matrix $R_{n}^{c}$, giving the rotation from the global coordinate system to the camera one, and a translation vector $t_{n}^{c}$ giving the translation from the origin of the global coordinate system to the one in the camera coordinate system C. The output of the “Vision” part in Figure 3 are $R_{c}^{n}\left(t\right)$, the rotation from the *Camera* frame to the *Navigation* frame at time (t) and $t_{c}^{n}\left(t\right)$, the translation from *Camera* frame to the *Navigation* frame $t_{c}^{n}\left(t\right)$ with a dimensionless scale factor s(t) at time (t).

For the relative pose estimation between images, feature points $x_{i}$ are extracted by SURF detector [43]. A sparse feature points extraction is operated to improve the reliability by suppressing redundantly extracted points. Then, extracted feature points in consecutive images are matched using the Sum Square Difference (SSD) distance to select unique correspondences. A filtering stage, using a geometric constraint such as epipolar constraint, is then applied with the M-estimator SAmple Consensus (MSAC) algorithm [44] to exclude outliers. A triangulation stage is operated only on inliers that are visible in the two images at times (t−1) and (t−2), and in the current image at time (t) to estimate the 3D points $X_{n}$ in the *Navigation* frame. To perform pose estimation using a calibrated camera, PnP algorithms constitute one of the most suitable solutions [45]. The camera pose is computed using the following formulation:(2)$$\tilde{x}{}_{i}=K\left[\begin{array}{ccc} R_{n}^{c} & | & t_{n}^{c} \end{array}\right]{\tilde{X}}_{n}$$(3)$$t_{n}^{c}=-R_{n}^{c}C^{n}$$where $C^{n}$ is the position of the camera’s optical center in the *Navigation* frame. The successive positions of the camera’s optical center are used for the scale estimation. It should be noted that we did not use any map optimization, such as pose graph optimization and bundle adjustment typically used in visual SLAM because they are computationally expensive. We simply implemented monocular Visual Odometry and did not keep the map in the memory as Dead-Reckoning.

### 3.3. Digital Terrain Model and Handheld Height

To complete the position estimation on the horizontal plane, the position on the vertical axis is determined in urban environments using data contained in 3D Geographical Information System (GIS), i.e., the Digital Terrain Model (DTM) and an estimation of the hand height holding a mobile device with respect to the pedestrian height. The DTM is a set of points referenced in planimetry (X,Y) and altimetry (Z). With an interpolation method, this provides the elevation of the ground level relief in digital form [46]. In the context of the use of an augmented reality applications for mobility assistance, the screen of the equipment held in the hand is considered maintained at a height $h_{hand}$ to see information. In order to estimate the handheld device position on the vertical axis, the height of several pedestrians $h_{ped}$, as well as the screen center height $h_{hand}$ on which an augmented reality display would be proposed, were measured. These measurements are presented in Table 1.

Thus, knowing the distance between the center of the screen and the optical center of the camera, the height of the equipment held in the hand can be experimentally defined as a ratio of the pedestrian height:(4)$$h_{hand}∼0.9\times h_{ped}$$

It should be noted that the height variations $\Delta h_{hand}\left(t\right)$, i.e., the variations of the DTM, is used to estimate the scale factor of the displacement along the vertical axis in the following section. In addition, since the variations in the DTM are very small, displacements on the vertical axis could be ignored to simplify the scale-factor calculation.

### 3.4. Scale Determination

Because steps and images are not sampled at the same times, an interpolation is needed. Then using the knowledge of pedestrian step lengths and the frequencies of both the visual measurement and the pedestrian’s step, a linear interpolation is operated to determine the pedestrian displacement between the instants of two consecutive images, as illustrated in Figure 4. The scale of the VO during the step ${k+1}^{th}$ is computed by using the $k^{th}$ step length based on the interpolation.

The step length estimation $step_{k}$ provides the magnitude of the pedestrian displacements on the horizontal plane. With the assumption that the displacements of the handheld device are mainly on the horizontal plane, displacements on the vertical axis are not taken into account.

Therefore, to estimate the scale s(t), a comparison is made between $D_{VO}\left(t\right)$, i.e., the magnitude of the displacement of the camera’s optical center $C^{n}$ on the horizontal plane estimated by the VO at times (t−1) and (t), and $D_{step}\left(t\right)$, i.e., an interpolation at times (t−1) and (t) of the estimated pedestrian step length on the horizontal plane $step_{k}$ between the instants of step (k−1) and (k).(5)$$D_{VO}\left(t\right)=∥C^{n}\left(t\right)-C^{n}\left(t-1\right)∥$$
(6)$$D_{step}\left(t\right)=step_{k}\times \frac{\Delta t_{image}}{\Delta t_{step_{k}}}$$

The scale factor s(t) is then defined as:(7)$$s\left(t\right)=\frac{D_{step}\left(t\right)}{D_{VO}\left(t\right)}$$

Thus, at each relative pose estimated by the VO, the scale is computed and used to correct the displacement of the handheld device on the horizontal plane in the *Navigation* frame. The final outputs are $R_{c}^{n}\left(t\right)$, the rotation matrix from the *Camera* frame to the *Navigation* frame at time (t) and $t_{c}^{n}\left(t\right)$, the translation vector from the *Camera* frame to the *Navigation* frame at time (t).

It should be noted that during the whole process, several steps can fail. In the “PDR” part, there might be some miss or false step detections. In the “VO” part, there might be false correspondences between features even if the matching process should limit it.

### 3.5. Known Object Recognition-Based Pose Estimation

To punctually suppress the drift and correct the relative pose estimate based on the scaled monocular Visual Odometry, the detailed knowledge of existing sparse known objects contained in 3D GIS is used. A known object allows to estimate absolute pose when one is detected in video frames [47]. The mean positioning accuracy is claimed to be 25 cm on the horizontal plane when a known object was detected in video frames. It should be noted that the track estimated with the monocular VO is highly dependent on initialization and reinitializations. Therefore, an inaccuracy in the known object-based pose estimation results in a bias in the orientation estimate of the trajectory. As the last view before a known object detection loss is a grazing view, which is the most degraded and the less accurate case for the known object-based pose estimation, the absolute pose estimated when the pedestrian is in a static phase in front of a known object since a few seconds is preferred to re-initialize the monocular VO. That corresponds to the most accurate case for the known object-based pose estimation. Figure 5 presents an illustration of our global approach for continuous pose estimation with a handheld device in urban environments. It comprises three main stages:The Known Object Detection;The pose estimation;
(a)The Scaled Monocular Visual Odometry (relative pose estimate).(b)The Known Object-Based Pose Estimation (absolute pose estimate).The AR visualization.

## 4. Experiments

### 4.1. Hardware Setup

According to the hardware setup needed to develop the proposed approach, the handheld device used in the experiments is composed of a monocular camera and an IMU rigidly attached together, as illustrated in Figure 6.

This hardware configuration gives access to raw data without the filters commonly applied to mobile device signals. To synchronize the IMU signals and the monocular camera recordings, timestamps from the GPS receivers embedded in both devices were used. The details of the devices are as follows:

A Garmin camera “VIRB 30 Ultra” (https://virb.garmin.com), set up with a fixed focal length, was used for image acquisition. The resolution of the camera was 1920×1080 pixels and was chosen to correspond to a standard resolution of smartphone’s acquisition. The video was acquired at 60 Hz frame rate. According to the computational resources in the device, the video was down-sampled at 10 Hz to reduce the computation time. The image resolution could also be resized, but this has not been done for the implementation of our solution.

A dedicated platform named ULISS [48] was used for measurements. It comprises a tri-axis Inertial Measurement Unit and a tri-axis magnetometer sampled at 200 Hz, a barometer, a High Sensitivity GPS receiver and an antenna. They are all low-cost sensors similar to those embedded in mobile devices owned by pedestrians.

### 4.2. Digital Terrain Model

The DTM, used in the experiments, was computed by the French National Geographical Institute (IGN) (professionnels.ign.fr/mnt). The resolution of the mesh is 1 meter with a decimetric accuracy for the altitude [49]. Other DTM, provided by public data (e.g., Open Street Map, Google Earth, etc.) could also be used. In our implementation, DTM data are processed using the OBJ format.

### 4.3. Scenario

A 0.7 km walk, which includes passages with an open view and buildings with important specular reflections, was performed by three different people in urban environments with the acquisition device held in hand. Details of the different pedestrians are given in Table 2. The pedestrian activity was in *Texting* mode at all times, i.e., the pedestrian walked while looking at the screen of the handheld device.

### 4.4. Estimation of Reference Waypoints and Tracks

During acquisitions, pedestrians walked on absolute reference waypoints marked on the ground (i.e., with a crossed-out white circle) and made a stop of several seconds over them, as illustrated in Figure 7. Their locations were measured with a centimetric accuracy using a geodetic dual-frequency GNSS receiver (http://www.septentrio.com/) in differential mode. The starting and finishing positions of the acquisition were also determined using a differential GNSS solution.

To assess pedestrian track estimates, reference tracks need to be established during acquisitions. As described in Table 3, in some urban spaces, GNSS-based solutions were not accurate and not available all the time. When the pedestrian walks in an open environment, the standard deviation of the positioning error was less than one metre. When the pedestrian walks in urban canyons, the standard deviation of the positioning error was up to 21 m for differential GNSS and up to 30 m for standalone GPS. These results represent the difficulty of urban location using such sensors, and is far from the accuracy required for an assessment.

Thus, to estimate an accurate and continuous reference track, the PERSY (PEdestrian Reference SYstem) platform [50] was mounted on the foot, on the same side as the handheld device was used during the data acquisition, as illustrated in Figure 7. The PERSY platform outputs location relative to a starting position and claims to have a 0.3% positioning mean error of the distance traveled. Quasi-static phases of the acceleration and the magnetic field are used to mitigate inertial sensor errors. GNSS phase measurements are added to the strap-down EKF to improve the positioning accuracy.

Figure 8 presents the relative PERSY reference track (in green) and the absolute reference waypoints (in white). Environmental specificities that may introduce difficulties in the monocular VO are also identified (in red).

## 5. Evaluation

### 5.1. Activities Classification and Step Length Estimation

Figure 9a presents the results of the activity classification for a walk in urban environments. It can be observed that the walking phases with the device held in hand in *Texting* mode (in green), and the moments when the pedestrian is static on each of the different reference waypoints (in pink) are correctly classified. Figure 9b presents the step lengths estimated, thanks to the walking frequency analysis and the knowledge of the pedestrian’s height. It also can be observed that the step frequency and the step lengths are not constant during the walk. This is due to the fact that the pedestrian has to avoid others pedestrians, bikers or cars, and to wait at pedestrian crossings. When the pedestrian walks normally, an amplitude of 40 cm is observed for the step lengths estimation, i.e., mainly between 1 m and 1.4 m. This validates the fact that the use of an average step length is not appropriate for an accurate scale estimation.

Table 4 presents the median step length of each pedestrian, as well as the standard deviation of their step lengths.

### 5.2. Estimated Trajectory

To perform the proposed scaled monocular VO, the initial pose in the *Navigation* frame must be estimated. For this process, position from GNSS and orientation from the IMU [51] were used. Then, to punctually suppress the drift and reinitialize the pose in the *Navigation* frame, the known object-based pose estimation proposed in [52] was used.

Figure 10 presents the relative PERSY reference track (in green) and the track estimated with the scaled monocular VO (in blue). From these results, it is observed that the estimated track is close to the relative PERSY reference track. This means that the proposed approach, using adaptive pedestrian step lengths estimates, allows to correctly scaled the monocular VO and to estimate the localization of a device held in hand by pedestrian while walking in urban environments. A visualization of the position estimation is also available on Youtube (https://youtu.be/3bSFrtF2lwU).

To assess the performance of the proposed approach, comparisons are made between the positions estimated with the scaled monocular VO and the relative PERSY reference tracks. Figure 11 presents the “Horizontal Positioning Error” and Figure 12 presents the “Cumulative Distribution Function” of the positioning errors for the three different pedestrians.

Because the relative PERSY reference track is not perfect due to slight inaccuracies that accumulate during the walk, an assessment of the positioning accuracy is also proposed in comparison to absolute reference waypoints. Thus, Table 5 presents the positioning errors between the scaled monocular VO and the absolute reference waypoints before a known object-based reinitialization.

A comparison with a scaling of monocular visual odometry by fixed-step length is also proposed in Table 5. A fixed-step length is defined in relation to the size of each pedestrian. The walking speed is also defined as fixed for all phases where pedestrians walk, and as zero for phases where pedestrians are static. It is observed that the proposed solution allows a positioning error of 1.6–7.5% of the walked distance, where the use of a fixed-step length only allows a positioning error of 7.1–12.7% of the walked distance. This comparison highlights the benefit of the proposed solution compared to the use of a fixed-step length.

As a general assessment of the proposed approach, it can be observed that there are positioning errors during the walk. There are mainly due to the fact that the estimation of the 3D world points coordinates is not perfect in the monocular VO, particularly in urban canyons and open areas, where extracted feature points correspond to elements that are several hundred meters away from the camera. However, these errors can be corrected if a known object is detected. In addition, due to important specular reflections, the implemented monocular VO does not allow to accurately estimate the changes in the orientation. This results in an error in the orientation estimate, which increases the positioning error at the end of the walk. This could be solved by using more sophisticated VO frameworks such as [53].

It should be noted that when the pedestrian makes small displacements, the proposed approach estimates a smooth trajectory, whereas the use of an average step length would have distorted the scale and the position estimate. Figure 13 presents a focus on an area where the pedestrian makes small displacements. A visualization is proposed on the horizontal plane and in a 3D environment containing 3D models of the known object and buildings around it.

## 6. Conclusions

In the context of pedestrian navigation, urban environments constitute challenging areas for both localization and handheld device pose estimation. Accurate position and orientation estimation are even more challenging, using only low-cost sensors available in general public devices, i.e., monocular camera and Inertial Measurement Unit.

To address these challenges, we propose a general approach, based on monocular Visual Odometry, to continuously estimate the pose of a handheld device. Our approach does not require any connection or any deployment of new infrastructures. To solve the scale ambiguity and suppress the scale drift in monocular Visual Odometry, an adaptive pedestrian step lengths estimation is used for the displacement on the horizontal plane. To complete the estimation, a handheld equipment height model, with respect to the Digital Terrain Model contained in Geographical Information Systems, is used for the displacement on the vertical axis. In addition, known objects allow to correct the pose estimate and reset the monocular Visual Odometry when one is detected in video frames.

A long walk of about 0.7 km with an IMU, that integrates low-cost sensors, combined with a camera held in hand has been conducted by three different pedestrians in urban environments with sparse known objects. An assessment is conducted using absolute reference waypoints, whose coordinates have been precisely determined with a Differential GNSS solution in an off-line phase. The assessment is completed by comparing results to a relative reference track obtained with a foot-mounted INS aided by GNSS phase measurements. The proposed approach enables to estimate the pose of a handheld device in urban environments, which is needed for augmented reality applications. Furthermore, this also allows to accurately estimate the pedestrian displacements without any use of GNSS positioning, which strongly deteriorates in urban and indoor environments. A comparison is also proposed between the use of an adaptive step length and the use of a fixed-step length to scale the monocular visual odometry. The proposed solution allows to achieve a positioning error between 1.6% and 7.5% of the walked distance, and confirms the benefit of the proposed solution compared to the use of a fixed-step length.

We plan that part of future works will be dedicated to more accurately estimating the pose, by fusing the presented global approach with PDR and barometric height in a tightly coupling process, mainly to improve the pose estimate in case of handheld device’s fast rotational motions. The proposed global approach will also be extended to other known objects in order to reduce the distance between two absolute pose estimates.

## Figures and Tables

**Figure 1 sensors-19-00953-f001:**
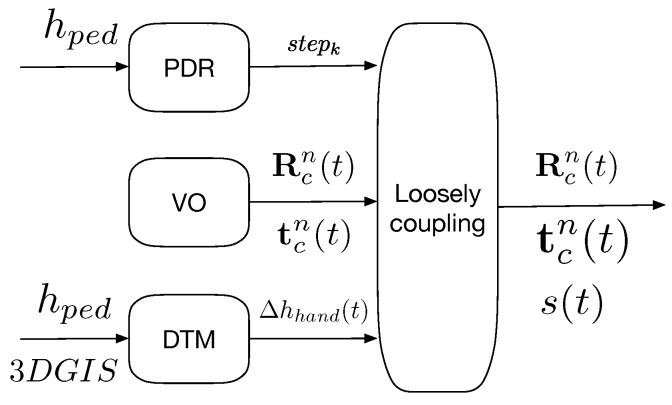
Block diagram of the proposed approach.

**Figure 2 sensors-19-00953-f002:**
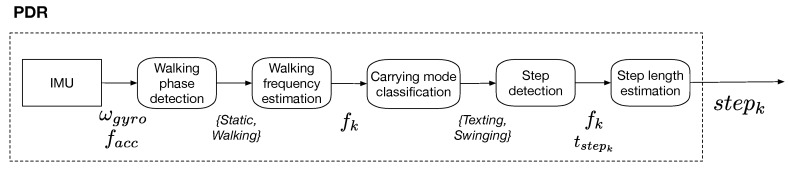
Detailed steps of “PDR” block.

**Figure 3 sensors-19-00953-f003:**
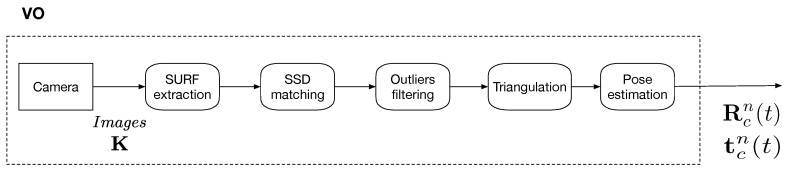
Detailed steps of “VO” block.

**Figure 4 sensors-19-00953-f004:**
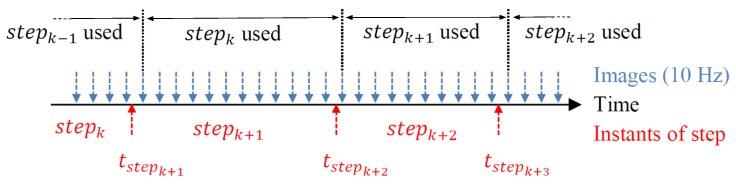
Step length interpolation.

**Figure 5 sensors-19-00953-f005:**
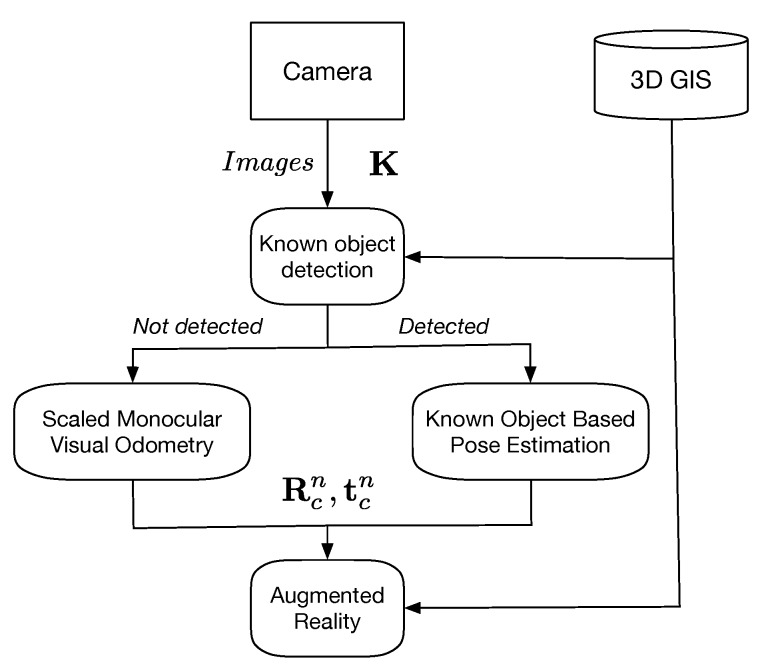
Global approach for continuous pose estimation in urban environments.

**Figure 6 sensors-19-00953-f006:**
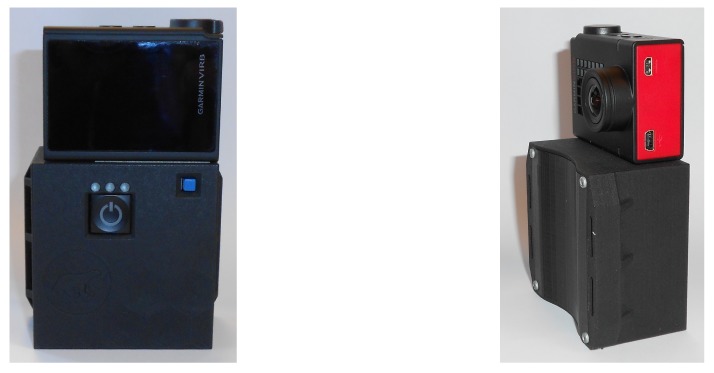
Hardware setup: the ULISS platform (**below**) and the Garmin VIRB camera (**top**).

**Figure 7 sensors-19-00953-f007:**
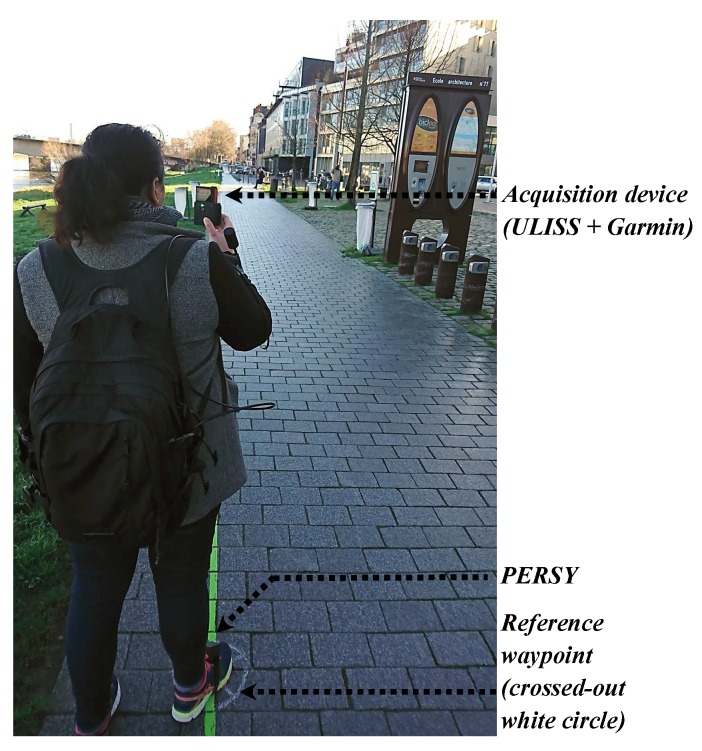
Handheld device (ULISS + Garmin), foot-mounted PERSY for reference track and reference waypoint.

**Figure 8 sensors-19-00953-f008:**
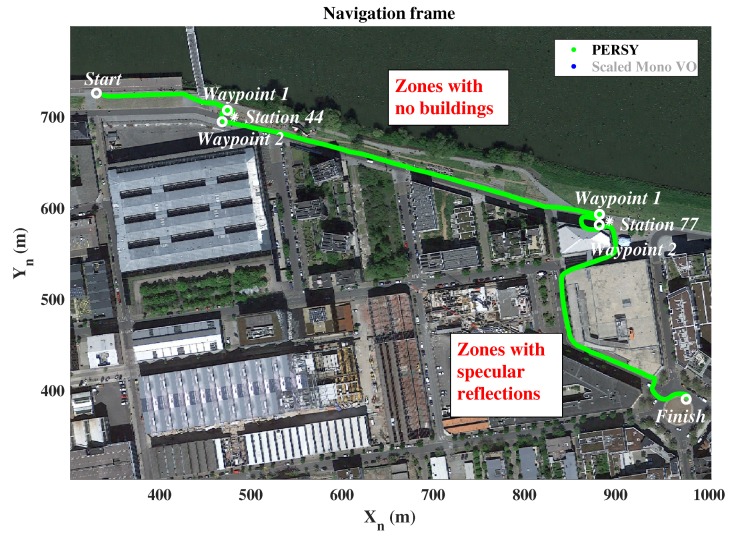
Absolute reference waypoints and relative PERSY reference track of the 0.7 km walk in urban environments.

**Figure 9 sensors-19-00953-f009:**
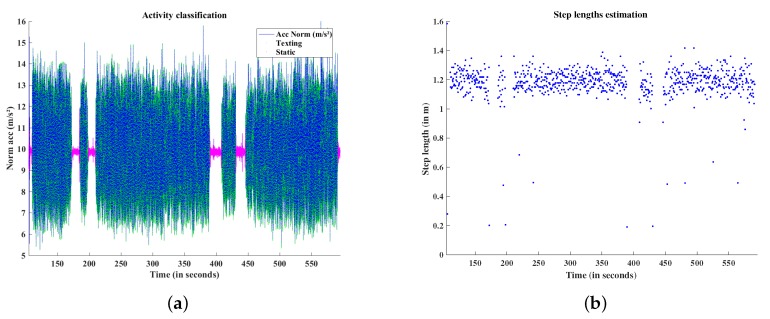
(**a**) Activities classification: *Static* (pink), *Texting* (green); (**b**) Step length estimation.

**Figure 10 sensors-19-00953-f010:**
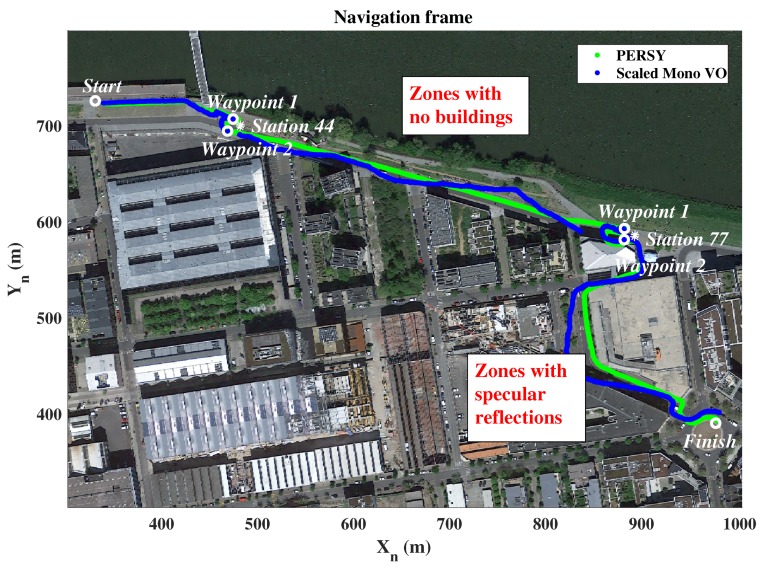
Scaled monocular Visual Odometry track, absolute reference waypoints and relative PERSY reference track of the 0.7 km walk in urban environments.

**Figure 11 sensors-19-00953-f011:**
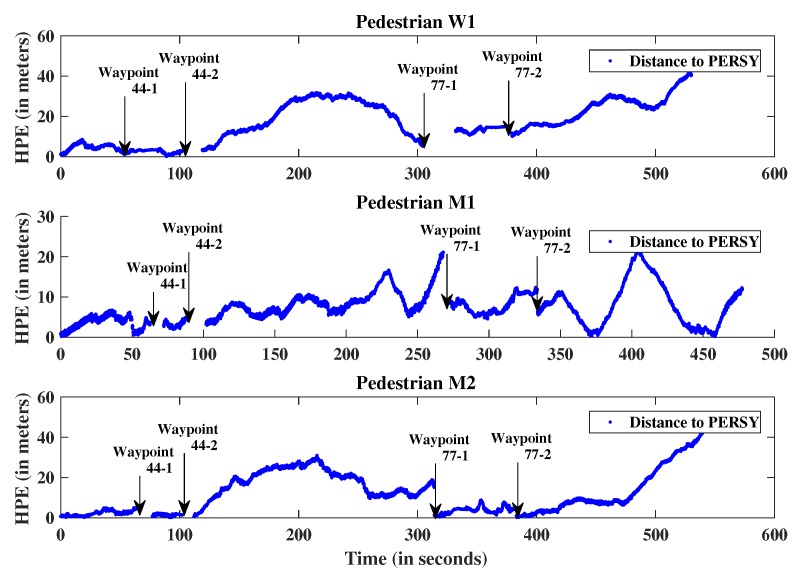
Scaled monocular Visual Odometry Horizontal Positioning Error compared to relative the PERSY reference track.

**Figure 12 sensors-19-00953-f012:**
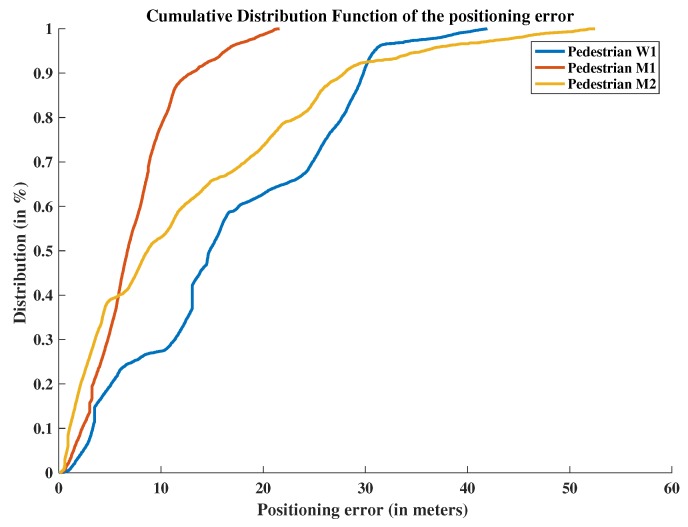
Scaled monocular Visual Odometry Cumulative Distribution function of the positioning errors compared to relative the PERSY reference track.

**Figure 13 sensors-19-00953-f013:**
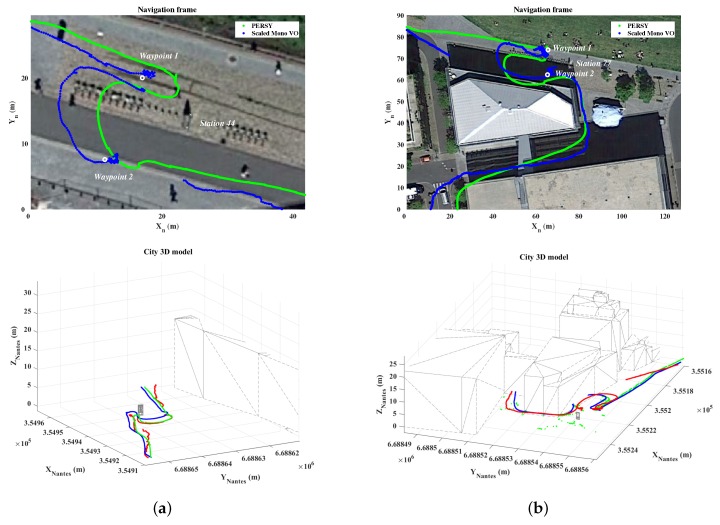
Example of a situation where the pedestrian makes small displacements: (**a**) vicinity of the station 44, (**b**) vicinity of the station 77.

**Table 1 sensors-19-00953-t001:** Measurements of pedestrian height and handheld equipment height.

Pedestrians	*M1*	*M2*	*W1*	*W2*	*W3*
Gender	Male	Male	Female	Female	Female
$h_{ped}$	1.87 m	1.80 m	1.69 m	1.69 m	1.60 m
$h_{hand}$	1.67 m	1.60 m	1.55 m	1.50 m	1.40 m
$h_{ped}-h_{hand}$	0.20 m	0.20 m	0.14 m	0.19 m	0.20 m
$h_{hand}/h_{ped}$	89%	88%	91%	88%	87%

**Table 2 sensors-19-00953-t002:** Acquisition measurements.

Pedestrians	*W1*	*M1*	*W2*
Gender	Female	Male	Male
$h_{ped}$	1.69 m	1.87 m	1.80 m
Acquisition duration	550 s	486 s	574 s

**Table 3 sensors-19-00953-t003:** GNSS measurements.

Pedestrians	*W1*	*M1*	*W2*
Differential GNSS positioning availability	90.7 %	94.5 %	81.2 %
Standalone GPS positioning availability	86.3 %	78.9 %	54.8 %

**Table 4 sensors-19-00953-t004:** Median step lengths and step lengths standard deviation.

Pedestrians	*W1*	*M1*	*W2*
Median step length	1.06 m	1.18 m	0.99 m
Step lengths standard deviation	0.20 m	0.14 m	0.29 m

**Table 5 sensors-19-00953-t005:** Scaled monocular Visual Odometry positioning errors and percentage of positioning error with respect to the walked distance compared to absolute reference waypoints and mean positioning error compared to relative the PERSY reference track. Results are given for the use of an adaptive step length (left) and for the use of a fixed-step length (right).

Pedestrians	*W1*	*M1*	*W2*
On waypoint 44-1 (after 120 m)	2.84 m | 27.97 m	3.58 m | 13.17 m	4.46 m | 35.16 m
	(2.3% | 23.3%)	(2.9% | 10.9%)	(3.7% | 29.3%)
On waypoint 77-1 (after 550 m)	6.60 m | 101.06 m	21.10 m | 78.74 m	15.11 m | 167.7 m
	(1.2% | 18.3%)	(3.8% | 14.3%)	(2.7% | 30.5%)
On finish (after 700 m)	22.43 m | 54.96 m	11.77 m | 49.80 m	52.56 m | 89.19 m
	(3.2% | 7.8%)	(1.6% | 7.1%)	(7.5% | 12.7%)
Mean positioning error	16.59 m | 26.58 m	7.33 m | 20.59 m	12.58 m | 45.09 m

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
