# Peer review of "Solving Monocular Visual Odometry Scale Factor with Adaptive Step Length Estimates for Pedestrians Using Handheld Devices"

_sensors, 2019, doi:10.3390/s19040953_

Reviewer 1 Report

This paper integrates data from inertial sensors, a monocular camera, and the height information from a DTM model for pedestrian localization. The topic and the innovation points mentioned by the authors are both interesting. However, the quality of the current version cannot meet the requirement for publication. Key information has been missed. Here are my comments:

1.As mentioned by the authors, one potential contribution of this paper is using adaptively estimated step length (SL) to aid the scale estimation in VO. However, I have not found any new things compared to the existing methods on step length estimation and VO scale estimation. More details are needed.

2.Additionally, for the above point, I doubt about whether the estimated SLareaccurate enough for VO scale estimation. As mentioned in Section 3.5, the accuracy for VO is 25 cm (in RMS, mean, or max value?). However, it is difficult for using handheld low-cost MEMS inertial sensors to achieve such accuracy. It seems that the contribution would be limited if one uses information that has a meter-level or decimeter-level accuracy to calibrate a variable that has a higher accuracy level.

3.As for adaptive SL estimation. How can it be “adaptive”? What type of information is needed for the estimation in real-time?

4.The use of DTM has not been described well. There are very limited descriptionsonthis point both theoretically and in the tests. Definitely more information is needed for the readers to understand this module. For example, how can we generate the DTM, in which format, what is the accuracy, how to use it, what is the computation load, etc.

5.The impact of device motion modes should be clarified. I understand that this paperis focusingon the handheld case. This is acceptable. However, from the method level, the impact of possible device motion changes should be described and tested. For example, this paper has an assumption that there is no displacementonthe vertical axis (above Figure 4). If there is a certain height change of the device, what would be the degradation to the method. It is necessary to quantify this. The accuracy of DTM would be significantly masked by the random change ofdevice heighton the human body.

6.The constraint “with handheld devices” should be reflected in the title.

7.A general description (and flow chart) is needed to show the relation between different modules, such as PDR, VO, DTM, initialization, scale estimation, etc.  The current Section 3 is not continuous. Fundamental knowledges and connections between modules aremissed.

8.The experimental verifications are far from qualified. The current version is more likely a simple PDR result report. I expected to see at least the following results: the comparison with existing method(s); the quantified error analysis. For example, what is the step-length estimation accuracy? What is its relation with VO estimation? Generation and use of DTM. The contributions from VO and DTM on the localization solution. Computational performance of the DTM method.

9.The test need more descriptions. For example, what are the sensor characteristics, what are the values for the parameters, how did the time synchronization between visual and inertialsensors beimplemented, etc.

Author Response

This paper integrates data from inertial sensors, a monocular camera, and the height information from a DTM model for pedestrian localization. The topic and the innovation points mentioned by the authors are both interesting. However, the quality of the current version cannot meet the requirement for publication. Key information has been missed. Here are my comments:

1.         As mentioned by the authors, one potential contribution of this paper is using adaptively estimated step length (SL) to aid the scale estimation in VO. However, I have not found any new things compared to the existing methods on step length estimation and VO scale estimation. More details are needed.

Answer: We propose to scale monocular visual odometry using real time adaptive step length estimation. Existing approaches are based on the use of a mean step length (fixed or sliding over time). In order to highlight our contribution, we have added a comparison showing the benefit of using an adaptive step length versus a fixed step length to solve the scale ambiguity.

2.         Additionally, for the above point, I doubt about whether the estimated SL are accurate enough for VO scale estimation. As mentioned in Section 3.5, the accuracy for VO is 25 cm (in RMS, mean, or max value?). However, it is difficult for using handheld low-cost MEMS inertial sensors to achieve such accuracy. It seems that the contribution would be limited if one uses information that has a meter-level or decimeter-level accuracy to calibrate a variable that has a higher accuracy level.

 Answer: The performances of the adaptive step length estimation, which is detailed in the references cited in section 3.1, gives an accuracy about 2-5% of the walked distance. This information has been added to the section 3.1 with the following sentence:

"It is claimed that this step length estimation is performed with a 2.5% up to 5% error on the walked distance."

To complete the paper, the errors on the estimated distance with respect to the reference walked distance have been computed and added in Table 5. This validates the fact that the accuracy of adaptive step length estimates is sufficient for solving the scale ambiguity of monocular visual odometry.

In section 3.5, the 25 cm accuracy corresponds to the mean positioning error in the horizontal plane obtained by known object recognition. The known object recognition is a process independent of the mono VO chosen and used to correct its drift.

To clarify this in the paper, the sentence "The positioning accuracy is claimed to be better than 25 cm in the horizontal plane when a known object was detected in video frames." was turn into: "The mean positioning accuracy is claimed to be 25 cm in the horizontal plane when a known object was detected in video frames."

3.         As for adaptive SL estimation. How can it be “adaptive”? What type of information is needed for the estimation in real-time?

 Answer: The step length is estimated using the pedestrian's height (that is fixed) and the real-time walking frequency estimate. The latter changes with walking velocity. Thus, the adaptive part comes from the varying step frequency estimation and the global locomotion state estimate (walking, static, etc.). This point is described in section 3.1.

4.         The use of DTM has not been described well. There are very limited descriptions on this point both theoretically and in the tests. Definitely more information is needed for the readers to understand this module. For example, how can we generate the DTM, in which format, what is the accuracy, how to use it, what is the computation load, etc.

Answer: A description has been added for the DTM in the theoretical part and a reference has been added, which details the methods for generating DTMs.

"The DTM is a set of points referenced in planimetry (X,Y) and altimetry (Z). With an interpolation method, this provides the elevation of the ground level relief in digital form [ref]."

The DTM is used to estimate the scale factor of the displacement along the vertical axis. To ease the reading and clarify the process, the order of sections 3.3 and 3.4 has been reversed. Furthermore, to help understanding, the following sentences have been added:

"It should be noted that the height variations , i.e. the variations of the DTM, is used to estimate the scale factor of the displacement along the vertical axis in the following section. In addition, since the variations in the DTM are very small, displacements on the vertical axis could be ignored to simplify the scale-factor calculation."

A subsection has been added in the paper to better detail the DTM in use.

"The DTM, used in the experiments, was computed by the French National Geographical Institute (IGN) (link: professionnels.ign.fr/mnt). The resolution of the mesh is 1 meter with a decimetric accuracy for the altitude [ref]. Other DTM, provided by public data (e.g. Open Street Map, Google Earth, etc.) could also be used. In our implementation, DTM data are processed using the OBJ format."

Because the computation was made in post-processing, loading and computation times were not measured. This will be part of future work, especially for real time implementation.

5.         The impact of device motion modes should be clarified. I understand that this paper is focusing on the handheld case. This is acceptable. However, from the method level, the impact of possible device motion changes should be described and tested. For example, this paper has an assumption that there is no displacement on the vertical axis (above Figure 4). If there is a certain height change of the device, what would be the degradation to the method. It is necessary to quantify this. The accuracy of DTM would be significantly masked by the random change of device height on the human body.

 Answer: We have placed the pedestrian in a context where he looks at his environment through the screen of his handheld equipment. Contexts where the pedestrian is no longer in texting mode and no longer looks at the screen of his handheld equipment, e.g. phoning or swinging mode, are not considered in the paper. Indeed, in this context, the monocular visual odometry is not unusable (e.g. camera against the arm or against the pants). They are out of the paper’s scope. However, these changes are important for general research on pedestrian navigation. Many researches have been published on how to detect those. These modes would require complementary solutions to address them.

In the paper, the assumption was made that there is no hand movement along the vertical axis (i.e. the height of the handheld device above ground is considered fixed). This hypothesis is realistic for VO or Augmented Reality use cases. In this context, the only variations considered along the vertical axis are those due to DTM variations. We are aware that this is a strong hypothesis and will ensure that its impact is quantified in future work.

6.         The constraint “with handheld devices” should be reflected in the title.

 Answer: The title was changed to: "Solving Monocular Visual Odometry Scale Factor with Adaptive Step Length Estimates for Pedestrians using Handheld Devices"

7.         A general description (and flow chart) is needed to show the relation between different modules, such as PDR, VO, DTM, initialization, scale estimation, etc.  The current Section 3 is not continuous. Fundamental knowledges and connections between modules are missed.

 Answer: Figure 1 has been redesigned to reflect this, and "Vision" was changed to "VO" in Figure 3.

8.         The experimental verifications are far from qualified. The current version is more likely a simple PDR result report. I expected to see at least the following results: the comparison with existing method(s); the quantified error analysis. For example, what is the step-length estimation accuracy? What is its relation with VO estimation? Generation and use of DTM. The contributions from VO and DTM on the localization solution. Computational performance of the DTM method.

 Answer: Table 4 has been added to provide results on adaptive step length estimation.  Questions regarding the step length estimation accuracy were also answered in question 2.

A comparison between the use of adaptive step length estimates and the use of an average step length has also been added in Table 5. The comparison highlights the benefit of our method compared to the use of a fixed step length.

The relationship between VO and step length estimation is detailed in sections 3.2 and 3.4.

Questions regarding the DTM were answered in question 4. However, since the reference is only available in the horizontal plane, it was not possible to assess the benefit of using DTM vertical variations to estimate the scale factor along the vertical axis.

9.         The test need more descriptions. For example, what are the sensor characteristics, what are the values for the parameters, how did the time synchronization between visual and inertial sensors be implemented, etc.

 Answer: Details concerning the characteristics of inertial sensors and their calibration are given in the references cited in sections 4.1 and 4.3.

Many thanks for your comment. Synchronization between the camera and the inertial data is very important to propose adequate signal processing filters. Time stamping is performed by the GPS receivers, embedded in all equipment. The data synchronization is guaranteed by the GPS time stamps.

To clarify this in the paper, we added the following sentence:

"To synchronize the IMU signals and the monocular camera recordings, timestamps from the GPS receivers embedded in both devices were used."

Reviewer 2 Report

This paper is a meaningful and interesting topic. However, there are some errors that should be avoided in high quality journal. The authors should proofread before submission. It will be better if the author can put emphasis on main academic contribution.

1. In the summary it is necessary to mention some phrase related to the results and conclusions obtained (over the line 12).

2. The paper does not make a definition of what SLAM means. There are no references to this concept (line 41).

3. In general, the authors adequately described the materials and methods of this study.

4. A digital elevation model is mentioned. However, it does not present the characteristics that it has for the implementation of the methodology (X, Y, resolution parameters) (lines 212 to 215)

Author Response

This paper is a meaningful and interesting topic. However, there are some errors that should be avoided in high quality journal. The authors should proofread before submission. It will be better if the author can put emphasis on main academic contribution.

1. In the summary it is necessary to mention some phrase related to the results and conclusions obtained (over the line 12).

Answer: Results obtained and analyses have been added to the summary.

2. The paper does not make a definition of what SLAM means. There are no references to this concept (line 41).

Answer: The definition of SLAM and a reference have been added.

"as is the case with Simultaneous Localization And Mapping (SLAM) techniques [3 refs]"

3. In general, the authors adequately described the materials and methods of this study.

Answer: Many thanks for your positive feedback.

4. A digital elevation model is mentioned. However, it does not present the characteristics that it has for the implementation of the methodology (X, Y, resolution parameters) (lines 212 to 215)

Answer: A description has been added for the DTM in the theoretical part and a reference has been added, which details the methods for generating DTMs.

"The DTM is a set of points referenced in planimetry (X,Y) and altimetry (Z). With an interpolation method, this provides the elevation of the ground level relief in digital form [ref]."

The DTM is used to estimate the scale factor of the displacement along the vertical axis. To ease the reading and clarify the process, the order of sections 3.3 and 3.4 has been reversed. Furthermore, to help understanding, the following sentences have been added:

"It should be noted that the height variations , i.e. the variations of the DTM, is used to estimate the scale factor of the displacement along the vertical axis in the following section. In addition, since the variations in the DTM are very small, displacements on the vertical axis could be ignored to simplify the scale-factor calculation."

A subsection has been added in the paper to better detail the DTM in use.

"The DTM, used in the experiments, was computed by the French National Geographical Institute (IGN) (link: professionnels.ign.fr/mnt). The resolution of the mesh is 1 meter with a decimetric accuracy for the altitude [ref]. Other DTM, provided by public data (e.g. Open Street Map, Google Earth, etc.) could also be used. In our implementation, DTM data are processed using the OBJ format."

Round  2

Reviewer 1 Report

The authors have answered all my questions properly. I do not have further comments. I suggest an acceptance of this paper.